# Immunomodulatory Activities of *Carica papaya* L. Leaf Juice in a Non-Lethal, Symptomatic Dengue Mouse Model

**DOI:** 10.3390/pathogens10050501

**Published:** 2021-04-21

**Authors:** Mohd Ridzuan Mohd Abd Razak, Nor Azrina Norahmad, Nur Hana Md Jelas, Adlin Afzan, Norazlan Mohmad Misnan, Adiratna Mat Ripen, Ravindran Thayan, Murizal Zainol, Ami Fazlin Syed Mohamed

**Affiliations:** 1Herbal Medicine Research Centre, Institute for Medical Research, National Institute of Health, Ministry of Health Malaysia, Shah Alam 40170, Malaysia; azrina.n@moh.gov.my (N.A.N.); nurhana.mj@moh.gov.my (N.H.M.J.); adlinafzan@moh.gov.my (A.A.); norazlan.misnan@moh.gov.my (N.M.M.); murizal@moh.gov.my (M.Z.); ami@moh.gov.my (A.F.S.M.); 2Allergy and Immunology Research Centre, Institute for Medical Research, National Institute of Health, Ministry of Health Malaysia, Shah Alam 40170, Malaysia; adiratna@moh.gov.my; 3Infectious Disease Research Centre, Institute for Medical Research, National Institute of Health, Ministry of Health Malaysia, Shah Alam 40170, Malaysia; ravin@moh.gov.my

**Keywords:** *Carica papaya*, leaf, dengue, virus, mouse, AG129, cytokine, RNA, juice

## Abstract

The role of *Carica papaya* L. leaf juice in immune dysregulation caused by dengue virus infection remains unclear. This study aimed to investigate the immunomodulatory activities of the freeze-dried *C. papaya* leaf juice (FCPLJ) on AG129 mice infected with a clinical DENV-2 (DMOF015) isolate. The infected AG129 mice were orally treated with 500 and 1000 mg/kg/day of FCPLJ, for three days. Platelet, leukocyte, lymphocyte and neutrophil counts were microscopically determined. The level of plasma proinflammatory cytokines was measured by multiplex immunoassay. The levels of intracellular cytokines and viral RNA were determined by RT-qPCR technique. The results showed that the FCPLJ treatment increased the total white blood cell and neutrophil counts in the infected mice. The FCPLJ treatment decreased the level of GM-CSF, GRO-alpha, IL-1 beta, IL-6, MCP-1 and MIP-1 beta in the plasma of the infected mice. The intracellular IL-6 and viral RNA levels in the liver of infected mice were decreased by the FCPLJ treatment. In conclusion, this study supports the potential immunomodulatory role of the FCPLJ in a non-lethal, symptomatic dengue mouse model. Further studies on the action mechanism of the *C. papaya* leaf juice and its possible use as adjunctive dengue immunotherapy are warranted.

## 1. Introduction

Dengue is a mosquito-borne disease, which is still endemic in more than 100 countries, including the Americas, South-East Asia and Western Pacific regions. It was estimated that 390 million dengue cases will occur each year, and 3.9 billion people in 128 countries are at risk of dengue virus infection [1]. The disease can cause a severe flu-like illness. Depending on the type of infection (primary or secondary) and strain of dengue virus, the condition of dengue illness can lead to lethal complication if left without proper medical management [1]. Furthermore, specific antiviral drug for dengue is still unavailable, as none of the potential dengue antiviral candidates tested, to date, were clinically effective [2].

The pathogenesis and severity of dengue are associated with immune dysregulation caused by dengue virus infection. For instance, the increases of specific proinflammatory cytokine and chemokine levels have been demonstrated in the serum of dengue patients at different phases of illness and clinical symptoms [3,4,5,6]. Therefore, searching for new anti-dengue candidates that can modulate immune dysregulation during dengue virus infection would be one of the best approaches in combating the pathogenesis of the disease.

Medicinal plants are a potential source of immunomodulatory agents [7]. One of the potential plant candidates is *Carica papaya* L., commonly known as papaya tree, which belongs to the family of Caricaceae. Traditionally, the juice or extract of *C. papaya* L. leaf is used to treat ailments such as fever caused by dengue, pyrexia, diabetes, gonorrhea, syphilis, inflammation and infected wounds [8,9,10]. The leaf of *C. papaya* L. has been extensively studied clinically for its thrombocytosis effect [11,12,13,14]. In addition, the infection induced cytokine modulating effect of *C. papaya* L. leaf extract has been reported in a pilot clinical study conducted on dengue patients [15]. This has highlighted the importance of further study focusing on immunomodulatory potential of *C. papaya* L. leaf during dengue virus infection. The immunomodulatory activities of *C. papaya* L. leaf juice and extracts have been proven by its ability to modulate cytokine production, enhance phagocytic activity and increase splenocyte proliferation [16,17,18]; however, the experiments were only limited to in vitro and in vivo models, which were not related to dengue disease. Therefore, the effect of *C. papaya* L. leaf juice on the immune dysregulation caused by dengue virus infection needs to be further studied in a proper disease model.

Establishing a dengue in vivo model that mimics dengue infection in human is challenging in drug discovery and vaccine preclinical studies, due to limited viremia and poor clinical outcomes [19]. Fortunately, the reliability of AG129 mouse, which lacks interferon signaling, as a preclinical model for dengue study has been proven [19,20]. AG129 mice inoculated with different strains of dengue virus could produce clinical signs similar to human dengue infections, such as viremia, thrombocytopenia, splenomegaly, systemic cytokine responses and plasma leakage, which lead to a severe form of the infection [21,22,23,24,25].

Our previous study has showed that *C. papaya* L. leaf juice treatment could affect the production of proinflammatory cytokines in the plasma and the liver of AG129 mice infected with a laboratory strain of dengue virus (New Guinea C) [26]. However, the infection with laboratory strain dengue virus was less symptomatic and only elicited small number of proinflammatory cytokines. The infection of AG129 mice with clinical dengue virus isolate has been shown to be symptomatic followed by the production of clinically relevant proinflammatory cytokines [21,22,23,24]. Therefore, this study was conducted to highlight the immunomodulatory activities of *C. papaya* L. leaf juice in a non-lethal but symptomatic dengue mouse model, which was established by the infection of AG129 mice with Malaysian clinical DENV-2 (DMOF015) isolate. The effect of freeze-dried *C. papaya* leaf juice (FCPLJ) treatment on viral RNA level, blood cell count, plasma cytokines production and intracellular cytokine levels in specific organs, such as the liver, spleen, kidney and brain were assessed in this study.

## 2. Results

### 2.1. Freeze-Dried C. papaya L. Leaf Juice Content

For quality control assessment, we quantified five of the main compounds (manghaslin, clitorin, rutin, nicotiflorin and carpaine) by liquid chromatography–mass spectrometry. The identification of these compounds in the FCPLJ was confirmed based on the comparison of their molecular formula and MS/MS fragmentations data with the literature. The representative chromatogram and the extracted ion chromatogram of each compound in the FCPLJ are presented in Appendix A. The quantitative analysis showed that the clitorin and manghaslin contents were the highest, i.e., 6.80 ± 1.23 mg/g and 5.71 ± 0.54 mg/g, respectively (Table 1). The carpaine content was 3.82 ± 0.34 mg/g (Table 1). The rutin and nicotiflorin contents were 1.46 ± 0.08 mg/g and 1.44 ± 0.24 mg/g, respectively (Table 1).

### 2.2. The Effect of FCPLJ Treatment on the Morbidity Level of AG129 Mice Infected with Dengue Virus

In the first phase of the experiment, the AG129 mice were intraperitoneally inoculated with 2 × 10^5^ PFU of clinical DENV-2 (DMOF015) isolate on day 0 and the bodyweight changes were monitored daily for 10 days after the infection. In addition, the treated mice were given 500 and 1000 mg/kg/day through oral route starting on day one to day three post-infection.

The bodyweight of the infected mice decreased from day three to day five post-infection (Figure 1). Then, the bodyweight of the infected mice started to increase on day six and fully recovered on day 10 post-infection (Figure 1). Treatment with 500 and 1000 mg/kg of FCPLJ did not affect the bodyweight changes of the infected mice (Figure 1). The mock infected mice and the FCPLJ treated mock infected mice did not show any changes in their bodyweight throughout the 10 days monitoring period. As the infection was not lethal, no mouse was found dead during the study.

In the second phase of the experiment, the efficacy of the FCPLJ was assessed on day four post-infection. The AG129 mice were intraperitoneally inoculated with 2 × 10^5^ PFU of clinical DENV-2 (DMOF015) isolate and followed by the FCPLJ oral treatment on day one to day three post-infection. On day four post-infection, the blood samples were collected via submandibular vein and the mice were euthanized for organ collection. Based on the organs weight, a significant spleen enlargement (*p* < 0.05) was observed in the infected mice (Figure 1) showing the sign of splenomegaly. There were no significant changes observed in the weight of other organs, such as liver, brain, kidney, testis, heart and lung (Appendix A). The FCPLJ treatment did not affect the splenomegaly development in the infected mice (Figure 1).

### 2.3. The Effect of FCPLJ Treatment on Blood Parameters

The blood samples collected on day four post-infection were processed for blood films and plasma isolation. The platelet, white blood cell, neutrophil and lymphocyte counts were microscopically determined, using the giemsa-stained blood films.

As compared to the mock-infected mice, the total white blood cell and neutrophil counts in the infected mice were significantly increased by 1.72- and 8.91-fold (*p* < 0.05), respectively (Figure 2). However, the lymphocyte and platelet counts were not significantly affected by the infection (Figure 2).

Treatment of 1000 mg/kg bodyweight (BW) FCPLJ significantly increased the total white blood cell and neutrophil counts by 1.44-fold (*p* < 0.05), as compared to infected mice without treatment (Figure 2). The levels of lymphocyte and platelet in the infected mice treated with FCPLJ were comparable to those of the mock-infected mice (Figure 2). The treatment of 1000 mg/kg FCPLJ has no effect on the platelet, total white blood cell, neutrophil and lymphocyte counts in the mock-infected mice (Figure 2).

### 2.4. The Effect of FCPLJ Treatment on Plasma Proinflammatory Cytokines

The level of 20 proinflammatory cytokines in the plasma collected on day four post-infection was measured by multiplex immunoassay. Out of 20 cytokines, three cytokines, interleukin-2 (IL-2), interleukin-4 (IL-4) and interleukin-12p70 (IL-12p70) were excluded from this study, due to low detection limit. The level of 14 cytokines was significantly increased (*p* < 0.05) in dengue-virus-infected mice (Figure 3 and Appendix A). Three cytokines, namely interleukin-5 (IL-5), interleukin-13 (IL-13) and macrophage inflammatory protein-2 (MIP-2), were not significantly affected by the infection (Appendix A).

Treatment with 500 and 1000 mg/kg FCPLJ significantly decreased the level of five cytokines (*p* < 0.05): Granulocyte-macrophage colony-stimulating factor (GM-CSF), growth-regulated protein alpha (GRO-alpha), interleukin-6 (IL-6), monocyte chemoattractant protein-1 (MCP-1) and macrophage inflammatory protein-1 beta (MIP-1 beta), in the infected mice (Figure 3). A significant decrease in IL-1 beta production (*p* < 0.05) was detected in the infected mice treated with 1000 mg/kg FCPLJ (Figure 3). The level of plasma cytokines in mock-infected mice treated with 1000 mg/kg FCPLJ was comparable with mock-infected mice without treatment (Figure 3 and Appendix A).

### 2.5. The Effects of FCPLJ Treatment on the Organs Intracellular Cytokines

The expression of intracellular cytokines was determined from total RNAs extracted from the organ’s tissues harvested on day four post-infection. The expression of four targeted cytokines, namely MCP-1, IL-6, interferon (IFN) gamma and tumour necrosis factor (TNF) alpha, in the liver, kidney, spleen and brain tissues was measured by quantitative RT-PCR. As compared to mock-infected mice, dengue-virus-infected mice significantly expressed (*p* < 0.05) a higher level of MCP-1 and IL-6 in the liver (9-fold and 22-fold, respectively) tissue (Figure 4). In addition, the IFN gamma and TNF alpha in the kidney tissue of infected mice were significantly upregulated, as compared to mock-infected mice (*p* < 0.05) (Appendix A).

The dengue-virus-infected mice treated with 500 mg/kg of FCPLJ showed a significant downregulation of liver IL-6 (4-fold), as compared to mock-infected mice (*p* < 0.05) (Figure 4). However, the downregulation of liver IL-6 was not significant (*p* > 0.05) in the infected mice treated with 1000 mg/kg FCPLJ (Figure 4). In addition, treatment with 500 and 1000 mg/kg FCPLJ significantly downregulated MCP-1 (6-fold and 8-fold, respectively), IL-6 (16-fold), IFN gamma (7-fold) and TNF alpha (5-fold and 6-fold, respectively) in the kidney of infected mice, as compared to mock-infected mice (*p* < 0.05) (Appendix A). The expression level of MCP-1, IL-6, IFN gamma and TNF alpha in the spleen and brain of infected mice was not affected by the FCPLJ treatment (Appendix A). There were no significant changes in the expression level of MCP-1, IL-6, IFN gamma and TNF alpha in the organs of mock-infected mice treated with 1000 mg/kg FCPLJ, as compared to mock-infected mice (Figure 4 and Appendix A).

### 2.6. The Effect of FCPLJ Treatment on Dengue Viral RNA

The levels of viral RNA in the plasma, spleen, liver, brain and kidney collected on day four post-infection were determined by quantitative RT-PCR technique. The viral RNA was detected in all organs and plasma of infected mice (Figure 5). The treatment of 1000 mg/kg FCPLJ significantly decreased the liver viral RNA level (*p* < 0.05) in the infected mice (Figure 5). However, the viral RNA level in the plasma, spleen, brain and kidney of infected mice was not significantly affected by the FCPLJ treatments (Figure 5).

## 3. Discussion

The signs of symptomatic infection by the Malaysian clinical DENV-2 (DMOF015) isolate were observed by the bodyweight reduction, viremia, increase in the level of white blood cell, neutrophil, proinflammatory cytokines and splenomegaly. Dengue viral RNA was detected in the liver, spleen, brain and kidney of infected AG129 mice. Apart from these organs, other studies have detected dengue virus antigen in other organs of AG129 mice, such as the skin, lymph nodes, bone marrow, lung, thymus, stomach and intestine [27,28]. The dengue virus tropism in AG129 mice correlates with human dengue autopsy studies [28]. As compared to our previous dengue mouse model, which was established by a lab strain dengue virus infection [26], the current symptomatic dengue mouse model showed higher production of proinflammatory plasma cytokines. However, the infection was not lethal, as the bodyweight of infected AG129 mice was recovered on day 10 post-infection. The similar clinical patterns have been observed previously in a non-lethal but symptomatic dengue mouse model, AG129 mouse infected with DENV-3 strain D83-144, Thai clinical isolate [21]. The increases in total white blood cell and neutrophil counts, as observed in our study, were similar to a previous study conducted by Milligan et al., on DENV-1 strain Western Pacific 74 (WP 74) [22].

In our previous studies, by using Ultra Performance Liquid Chromatography Triple Time-Of-Flight Electrospray Interface Mass Spectrometry (UPLC-TripleTOF ESI-MS), we identified at least 12 compounds in *C. papaya* leaves consisting of phenolic acids (caffeic acid, *p*-coumaric acid, malic acid, quinic acid, caffeoyl malate, *p*-coumaroyl malate isomers and feruloyl malate isomers), piperidine alkaloid (carpaine) and glycosylated flavonol (clitorin, manghaslin, rutin and nicotiflorin) [29]. By using Ultrahigh-Performance Liquid Chromatography coupled with ESI Q-Exactive Orbitrap High-Resolution Accurate Mass Spectrometry (UHPLC–HRAMS), another 12 compounds were detected, namely fisetin, quercetin, kaempferol, citropten, myricetin, morin, chlorogenic acid, sinapsic acid, isoquercetin, astragalin, protocatechuic acid and dehydrocarpaine [30]. In addition, other compounds of *C. papaya* leaves such as catechin, hesperitin, dicoumarol, tocopherol, hydroxyflavanone, vanillic acid, linoleic acid and amino acids were also detected by other studies conducted in India [8,16]. The contents of manghaslin, clitorin, rutin and nicotiflorin in FCPLJ were comparable to the content of papaya leaves obtained in Kalimantan, Indonesia (Nugroho et al., 2017). Carpaine, the biological active marker for thrombocytosis, was 4–30-fold lower (3.82 ± 0.34 mg/g), as compared to papaya leaves collected in Gujarat, India (16.79–117.10 mg/g in various preparations) [31].

The potential immunomodulatory activity of *C. papaya* L. leaf juice was observed when the total white blood cell and neutrophil counts were increased in the infected AG129 mice treated with 1000 mg/kg of FCPLJ. The increase in white blood cell components by *C. papaya* L. leaf juice or extract treatments has been reported in previous studies on healthy rats [17], thrombocytopenic rats [16] and thrombocytopenic dengue patients [32,33,34]. The immunomodulatory potential of *C. papaya* leaf has been demonstrated clinically, where the reduction of proinflammatory cytokine level, such as IL-6, was observed in severe thrombocytopenic dengue patients treated with *C. papaya* leaf extract (Caripill) [15]. In addition, by multiplex cytokine screening, our study showed that FCPLJ treatment could reduce the level of GM-CSF, GRO-alpha, IL-6, MCP-1, MIP-1 beta and IL-1 beta in the plasma of dengue-virus-infected AG129 mice. The anti-inflammatory activity of FCPLJ could be due to the presence of flavonoids [16,17,29], such as quercetin, kaempferol and rutin, which have been previously studied for their abilities in reducing the production of proinflammatory cytokines [7,35]. The ability of FCPLJ to decrease the proinflammatory cytokines could possibly reduce the severity of dengue, as IL-6, MCP-1, IL-1 beta and MIP-1 beta have been associated with severe dengue infection [3,36,37,38,39].

The mechanism of *C. papaya* leaf in increasing the platelet level has been associated with thrombocytosis activity, as the megakaryocytes genes, arachidonate 12-lipoxygenase (ALOX-12) and platelet-activating factor receptor (PTAFR), were found to be upregulated in dengue patients receiving *C. papaya* leaf juice [14]. Another possible role of *C. papaya* leaf juice or extract is by affecting the platelet–leukocyte aggregation, which is a peripheral mechanism of thrombocytopenia in dengue [40]. This is because most of the FCPLJ-affected cytokines have been previously associated with leukocytes activation and recruitment during dengue virus infection (see Appendix A). In dengue infection, the increase of cytokines such as IL-6 and IL-1 beta could cause the activation of monocytes, platelets and coagulation enzymes, hence promoting the localization and interaction of monocytes–platelets at the site of infection, such as endothelial cells [40,41]. This mechanism was proposed as one of the factors that could cause thrombocytopenia in dengue patients [40,41,42]. Therefore, the ability of FCPLJ treatment to reduce the inflammatory cytokines might abrogate the monocytes–platelets aggregation and subsequently prevent thrombocytopenia. In addition, quercetin, one of the flavonoids found in FCPLJ [30], has been found to play a role in platelet aggregation inhibition activity [43,44]. However, the proposed mechanism needs to be validated further by a study on monocytes–platelets interaction with the presence of FCPLJ or immunomodulatory substances of FCPLJ.

The potential dengue antiviral activity of *C. papaya* leaf juice and extract has been highlighted in in vitro and clinical studies [8,15]. Briefly, *C. papaya* leaf extract was shown to decrease the expression of envelope and NS1 proteins in DENV-infected human monocyte cells [8]. In addition, the decrease in plasma NS1 level was reported in dengue patients receiving *C. papaya* leaf extract; however, the effect needs further validation in larger study population [15]. Meanwhile, in our study, the viral RNA level in the plasma of the infected AG129 mice was not affected by the FCPLJ treatment. Interestingly, the FCPLJ treatment only decreased the viral RNA level in the liver but not in other organs (spleen, brain and kidney). There is a possibility that some of the compound(s) in the FCPLJ becomes more active after being metabolized in the liver. However, further study is needed to investigate the involvement of FCPLJ’s metabolized product in the liver. Furthermore, downregulation of proinflammatory cytokine, IL-6, in the liver of infected mice treated with FCPLJ indicates that the FCPLJ could possibly modulate antiviral and anti-inflammatory effects in a specific organ, such as the liver. The association of dengue virus infection with the increase in the liver proinflammatory cytokines was highlighted in our previous study on dengue fever mouse model [26]. The treatment with 1000 mg/kg of FCPLJ has downregulated the expression of proinflammatory cytokines and receptors (CCL6, MCP-2, MCP-5, CCL17, IL1R1, IL1Ra, NAMPT and PF4) in the liver of infected mice [26]. Furthermore, the FCPLJ treatment has also downregulated the genes associated with the endothelial cell biology (ITGB3, ICAM1 and FN1) that are involved in the endothelial permeability process during dengue virus infection [45]. All of the evidences showed that the FCPLJ could be the hepatoprotective agent by modulating the cytokine induced inflammation, hence decreasing the dengue virus replication. The potential biological active compounds, quercetin and fisetin, flavonoids, detected in the FCPLJ might play their role in exerting the anti-inflammatory and antiviral activities in the liver during dengue virus infection. Quercetin and fisetin have been shown to inhibit dengue virus replication in vitro [46,47]. In silico study has also shown that the dengue viral NS2B-NS3 protease could be the potential target for quercetin [48]. Besides antiviral activity, both of the compounds showed in vitro anti-inflammatory activities by reducing the TNF alpha and IL-6 productions in human U937-DC-SIGN macrophages infected with dengue virus [49].

Our study has showed the potential immunomodulatory role of FCPLJ, based on the augmentation of total white blood cell and neutrophil counts, and a decrease in the proinflammatory cytokines level in the treatment group of dengue virus infected-AG129 mice, hence implicating the potential of FCPLJ to be used as adjunctive immunotherapy for dengue. However, there are several limitations that need to be highlighted. Our study could not further translate the affected parameters on the morbidity level, and plasma leakage as the infection was not lethal and hemorrhagic. Perhaps, a future study on severe dengue mouse model could further highlight the effect of FCPLJ on the plasma leakage development and survival level. Our dengue mouse model also demonstrated a slight reduction (not statistically significant) of platelet levels on day four post-infection. There is a possibility that thrombocytopenia effect could be more obvious at earlier day of infection such as on day two post-infection, which has been previously demonstrated by a study conducted by Sarathy et al. [21] on a symptomatic but non-lethal dengue mouse model. Therefore, a future study that observed the platelet level at more than a single time-point could highlight the kinetic of thrombocytopenia in dengue mouse model.

Preclinical and clinical studies have showed that the *C. papaya* leaf juice was able to prevent thrombocytopenia [11,14,16,31,50]. This could be due to the action of carpaine, an alkaloidal compound of *C. papaya* L. leaf juice, which was previously demonstrated to increase the platelet in busulfan induced thrombocytopenic Wistar rats [31]. In this study, healthy AG129 mice treated with high dose of FCPLJ (1000 mg/kg BW) did not show any increase in its basal platelet level, which was contradicted with other studies on wild-type murine and rodent strains [16,50,51,52]. This could be due to the difference of species, strains and disease model used in other studies. In addition, the different amount of carpaine between extracts could not be excluded, as the quantitative analysis has showed that the carpaine content in the FCPLJ used in this study was lower than the study reported by Zunjar et al. [31]. Hence, the quantities of compounds, including carpaine, that are biologically active need to be determined in order to produce the standardized *C. papaya* leaf juice or extract formulation in the future.

## 4. Materials and Methods

### 4.1. Experimental Animals and Husbandry

The study was approved by the Animal Care and Use Committee, Ministry of Health Malaysia (ACUC-MOH), ACUC/KKM/02(9/2016). All experiments were conducted in the Non-Clinical Research Facility, Laboratory Animal Research Unit, Special Resource Centre, Institute for Medical Research, Kuala Lumpur, Malaysia. All procedures on mice were conducted by trained personnel and supervised by a certified veterinarian.

The AG129 mice (129/Sv mice deficient in both alpha/beta and gamma interferon receptors) (male, 4 to 5 weeks old) were obtained from Marshall BioResources, United Kingdom. The mice were housed in individual ventilated cages, supplied with reverse osmosis drinking water and mouse pellet, ad libitum. The mice were exposed with artificial light, 12 h light and 12 h dark. The temperature range of experimental room was maintained within 22 to 26 °C. The mice were quarantined for 2 weeks and acclimatized for 7 days before the experiment. During the quarantine period, daily health and morbidity assessments were conducted by the veterinarian. The health assessment was continued during the acclimatization period. The experiment started when the age of the mice was between 7 and 8 weeks old (20–27 g of bodyweight).

### 4.2. Study Design

The experiment was divided into 2 phases. The first phase was the morbidity monitoring experiment, which involves the monitoring of bodyweight changes for a period of 10 days post-infection. The second phase was the efficacy evaluation experiment, which involves the analysis of viral RNA, blood components, plasma cytokine screening and intracellular cytokine expression on day 4 post-infection. Both experimental phases consisted of 5 experimental groups: mock infected (mock) (*n* = 5), mock infected with FCPLJ treatment at 1000 mg/kg (mock + FCPLJ 1000) (*n* = 5), infected without FCPLJ treatment (infected) (*n* = 5), infected with FCPLJ treatment at 500 mg/kg (infected + FCPLJ 500) (*n* = 5) and infected with FCPLJ treatment at 1000 mg/kg (infected + FCPLJ 1000) (*n* = 5) groups. The number of mice per group (*n* = 5) was determined by resource equation approach [53]. Random numbers for assignment of 5 animals per each group were generated by RAND function in the Excel, Microsoft Office software. To minimize the bias during the conduct of the experiment, all procedures, data collection, and analysis were done by a different group of study personals or individuals.

### 4.3. Dengue Virus Preparation and Inoculation

The Malaysian clinical dengue virus, serotype 2 or DENV-2 (DMOF015) was isolated from hemorrhagic dengue patient by Virology Unit, Infectious Disease Research Center, Institute for Medical Research, Malaysia. The dengue virus was propagated and prepared as mentioned previously [30]. The infection was performed by intraperitoneal inoculation of 2 × 10^5^ plaque forming unit (PFU) of DENV-2 (DMOF015) in a 250 µL media solution.

### 4.4. Freeze-Dried C. papaya L. Leaf Juice (FCPLJ) Preparation and Quantitative Analysis of Selected Markers

The FCPLJ was prepared from green leaves of organically grown *C. papaya* L. trees as mentioned previously [30]. The plant was authenticated by Ms. Tan Ai Lee, a botanist from the Forest Research Institute Malaysia (FRIM), Kepong, Malaysia. The voucher specimen (Voucher No: 007/10) was deposited in the FRIM. The characterization of the FCPLJ by liquid chromatography–mass spectrometry has confirmed the availability of 5 major compounds; carpaine (alkaloid), rutin, manghaslin, clitorin and nicotiflorin (flavonoids) [29,30] and the same batch of the characterized FCPLJ was used for dosing in this experiment.

The content of FCPLJ was quantified by full scan and selected ion monitoring of five compounds (clitorin, manghaslin, rutin, nicotiflorin and carpaine), on a Dionex Ultimate 3000 Series RS pump coupled with Q-Exactive Orbitrap mass spectrometer and ESI interface. Data were processed with Xcalibur software version XX (All Thermo Fisher Scientific, MA, USA). The electrospray ionization was operating in negative and positive ionization. For this analysis, rutin and nicotiflorin were purchased from Extrasynthese (Genay, France). Cynaroside was from Sigma-Aldrich (Madrid, Spain), Emetine hydrochloride from European Pharmacopoeia Reference Standard and carpaine from Chengdu Biopurify Phytochemical Ltd (Chengdu, China). Clitorin and manghaslin were isolated from *C. papaya* leaves, using an in-house method. The calibration standard curves were prepared in 10,000, 5000, 1000, 500, 100 and 50 ng/mL, containing a mixture of clitorin, manghaslin, rutin, nicotiflorin and carpaine in equal amounts. Emetine hydrochloride and cynaroside were used as internal standards (100 ng/mL). Separation of the five compounds was achieved on an Acquity UPLC^®^ BEH HSST3 (1.8 µm, 2.1 × 100 mm; WATERS) column, flow rate of 0.5 mL/min (40 °C) with water (A) and acetonitrile (B), both with 0.1% formic acid. The following gradient was used: 1 to 15% B from 0 to 3 min, 15–50% B from 3 to 6 min, 50–95% B from 6 to 9 min, isocratic step at 95% B for 1 min and equilibration step of 2 min. The FCPLJ samples (5 mg/mL in methanol) were analyzed in triplicates. Injection volume for all standards and samples were 1 µL.

### 4.5. Dosing

The doses were determined based on general toxicology and clinical studies of *C. papaya* L. leaf juice, as mentioned in our previous study [30].The correct amount of FCPLJ was dissolved in water, to get 500 and 1000 mg/kg BW of dosing regimens. The oral dosing was done once a day, starting from day 1 to day 3 post-infection, by using 22G feeding needles. The dosing volume was equivalent to 10 mL/kg bodyweight.

### 4.6. Morbidity and Clinical Observation

Upon infection, bodyweight changes in the phase 1 experiment (morbidity monitoring experiment) was monitored daily until day 10 post-infection. The signs of illness were monitored for both phase 1 and phase 2 experiment groups, once a day, and scored based on 1 to 5 scale: 1 = healthy; 2 = mild sign of lethargy and ruffled fur; 3 = intermediate level of lethargy, ruffled fur and hunched posture; 4 = very lethargy, ruffled fur, hunched posture and limited mobility; 5 = moribund with limited to no mobility and inability to reach food or water [54]. During the experiment, none of the mice meet the criteria for immediate euthanasia such as weight loss of more than 20% of initial body weight and/or moribund and/or paralyzed. At the end of the experiment, the surviving mice were euthanized, for sample collection. The euthanasia was performed in the fume hood, by the veterinarian, using open-drop exposure to 5% isoflurane.

### 4.7. Sample Collection

In the phase 2 experiment, the blood samples (0.2 mL) were collected into a K_3_EDTA Microtainer through submandibular vein, on day 4 post-infection. The blood samples were processed for thin blood film preparation and plasma collection [30]. Then, the mice were euthanized, using open-drop exposure to 5% isoflurane in the fume hood. The livers, kidneys, brains and spleens were harvested and weighed. The tissues of the collected organs were kept in RNA later solution, for preservation at −40 °C, prior to the total-RNA-extraction process.

### 4.8. Blood Counting

The platelet [55], total white blood cell (WBC) and blood differential [56] counts from the Giemsa stained blood smears were performed manually by using microscopy technique. Briefly, platelet count (platelet count/µL) was determined by multiplying the average platelet number of platelets in 10 oil immersion fields by 15,000/µL. The total WBC was determined by multiplying the average number of WBC (WBC count/µL) in 10 power fields (40×) by 2000/µL. Absolute neutrophil and lymphocyte counts (neutrophil or lymphocyte count/µL) were determined by multiplying the number of cell per 100 WBC (40× power fields) by total WBC count/µL.

### 4.9. Multiplex Cytokine Immunoassay

The plasma (50 µL) was used as neat in the cytokine screening, using the ProcartaPlex Multiplex Immunoassay kit Mouse 20-plex (Thermo Fisher Scientific, Waltham, MA, USA) according to the manufacturer’s instruction, on a Luminex platform. The 20 targeted cytokines were granulocyte-macrophage colony-stimulating factor (GM-CSF), interferon (IFN)-gamma, interleukin-1 beta (IL-1 beta), interleukin -12p70 (IL-12p70), interleukin-13 (IL-13), interleukin-18 (IL-18), interleukin-2 (IL-2), interleukin-4 (IL-4), interleukin-5 (IL-5), interleukin-6 (IL-6), tumour necrosis factor (TNF)-alpha, eotaxin, growth-regulated protein (GRO)-alpha, interferon gamma-induced protein-10 (IP-10), monocyte chemoattractant protein-1 (MCP-1), monocyte chemoattractant protein-3 (MCP-3), macrophage inflammatory protein-1 alpha (MIP-1 alpha), macrophage inflammatory protein-1 beta (MIP-1 beta), macrophage inflammatory protein-2 (MIP-2) and regulated upon activation, normal T cell expressed and presumably secreted (RANTES). The reading was measured, using xPONENT software version 4.2 (www.luminexcorp.com, accessed on 21 March 2021). The concentration (pg/mL) of protein was calculated, using MasterPlex QT version 2.0.0.59, based on the standard curve of each cytokine and chemokine.

### 4.10. Quantitative Reverse Transcription PCR (RT-qPCR)

The viral RNA was extracted from 40 µL of plasma by using QIAamp Viral RNA Mini kit (Qiagen, Hilden, Germany). The tissue’s total RNA was extracted by using RNeasy mini kit (Qiagen, Hilden, Germany). The concentration of total RNA was determined by using NanoDrop spectrophotometer (Thermo Fisher Scientific, Waltham, MA, USA). The viral RNA copy number in the plasma and organs’ tissues was determined by one-step RT-qPCR method (Applied Biosystems 7500 fast, Thermo Fisher Scientific, Waltham, MA, USA), using QuantiTect SYBR^®^ Green RT-PCR detection kit (Qiagen, Hilden, Germany). The primers and cycling temperatures for dengue viral RNA detection were as described by Chutinimitkul et. al. [57]. The purified dengue viral RNA with known copy number was used as a standard for copy-number determination. The viral RNA copy numbers in the tissues were normalized by total RNA concentration and to an endogenous housekeeping gene, GAPDH Ct value.

The cytokine expressions in the organ tissues were performed by RT-qPCR, using QuantiTect Probe RT-PCR detection kit (Qiagen, Hilden, Germany). The primers and cycling temperatures for the cytokine expression analysis were as mentioned by Overbergh et al. [58]. The level of expression was determined by comparative Ct method [59]. The Ct value of each samples was normalized to an endogenous housekeeping gene, GAPDH Ct value. All organ tissues from 5 mice (*n* = 5) in each group were analyzed for cytokine expression, except for the kidney tissues of the infected group, which were limited to 4 (*n* = 4), because one of the samples was found to be degraded.

### 4.11. Statistical Analysis

The mean difference between groups was analyzed by ANOVA, with multiple comparison test, using the GraphPad Prism software version 6.01. Any differences were considered significant when the *p*-value was less than 0.05 (*p* < 0.05).

## 5. Conclusions

Our study has highlighted the potential immunomodulatory roles of FCPLJ, such as the augmentation of total white blood cell and neutrophil, and anti-inflammatory activity in a non-lethal, symptomatic dengue mouse model. The FCPLJ treatment did not affect the viral RNA level in the plasma. Identification of FCPLJ’s biological active compound(s) is warranted for future studies on the action mechanisms and its possible use as an adjunctive immunotherapy for dengue patients.

## Figures and Tables

**Figure 1 pathogens-10-00501-f001:**
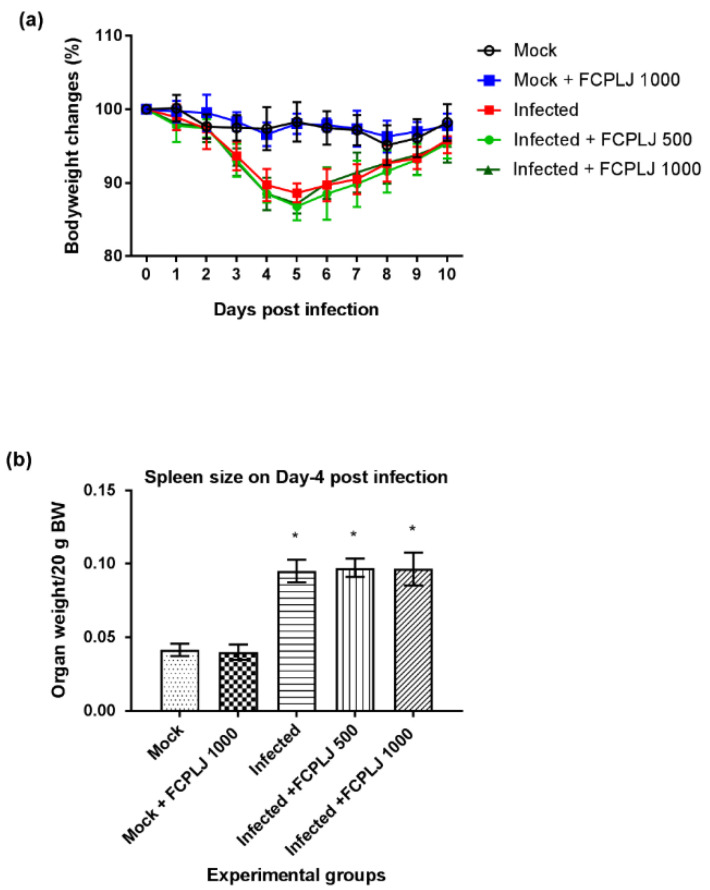
The morbidity level of AG129 mice during dengue virus infections. (**a**) The trends of bodyweight reduction and recovery of dengue virus infected mice and other experimental groups from day 0 to day 10 post-infection showed that the infection was symptomatic but not lethal. (**b**) The weight of the spleen of each experimental mouse was measured on day four post-infection. A splenomegaly was observed in the infected mice group. The FCPLJ treatment was not affecting the bodyweight changes and spleen size of infected mice. The bars represent the mean value of bodyweight changes in percentage or spleen weight per 20 g bodyweight (BW) ± standard deviation. The comparison between groups was analyzed by ANOVA, using Tukey’s multiple comparison test. The asterisk (*) represents significant difference (*p* < 0.05), as compared to mock-infected group. Each experimental group consists of five mice (*n* = 5).

**Figure 2 pathogens-10-00501-f002:**
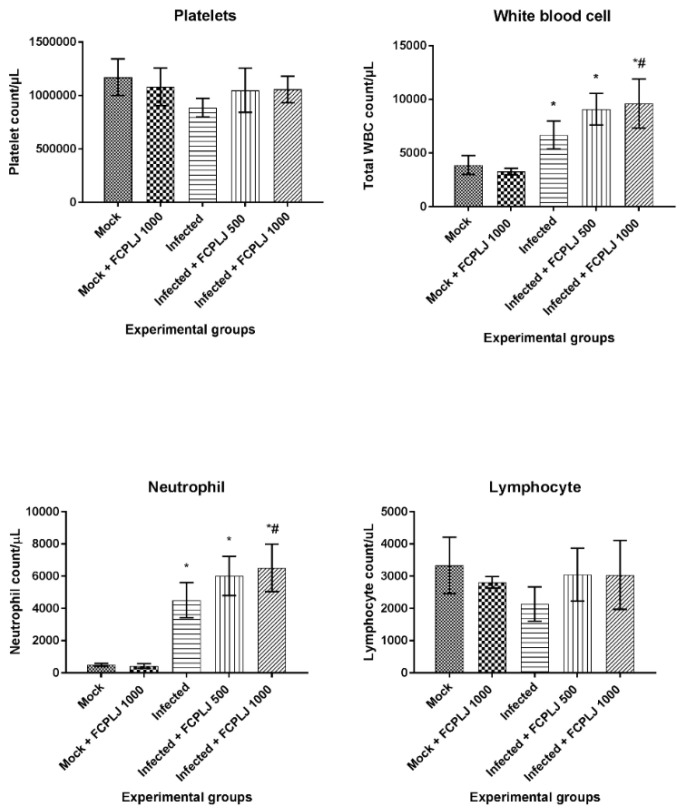
The level of platelet, total white blood cell, neutrophil and lymphocyte of AG129 mice experimental groups. The infected AG129 mice were treated orally with distilled water (infected) and FCPLJ (infected + FCPLJ 500 or 1000 mg/kg), for three days, at 24 h post-infection. The platelet, total white blood cell, neutrophil and lymphocyte counts were determined from giemsa-stained blood smear collected on day four post-infection. The bars represent the mean count ± standard deviation. The comparison between groups was analyzed by ANOVA, using Tukey’s multiple comparison test. Note: The asterisk (*) represents significant difference (*p* < 0.05) when compared with mock-infected and mock + FCPLJ 1000 mice. The hash (#) represents significant difference (*p* < 0.05) when compared with the infected mice. Each experimental group consists of five mice (*n* = 5).

**Figure 3 pathogens-10-00501-f003:**
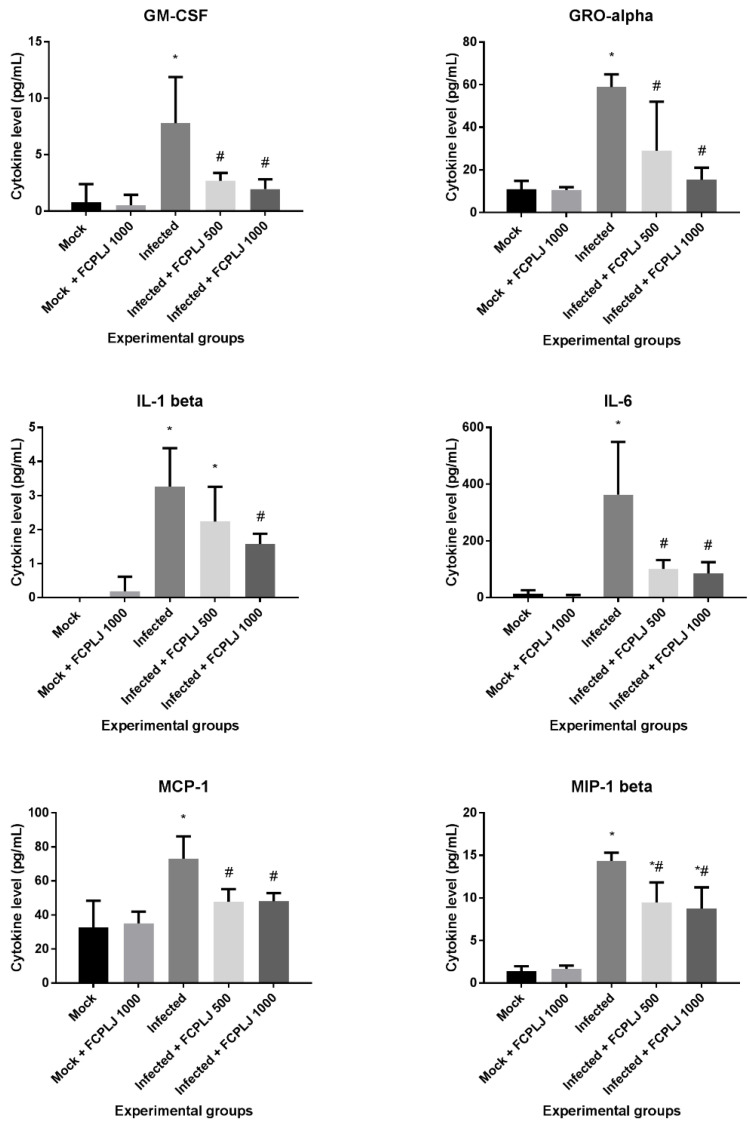
The level of plasma cytokines affected by the FCPLJ treatment. The cytokine levels in the plasma collected on day four post-infection were detected by multiplex assay system. Six plasma cytokines in infected AG129 mice were found to be significantly decreased (*p* < 0.05) upon treatment with either or both 500 and 1000 mg/kg FCPLJ. The bars represent the mean values ± standard deviation. The comparison between groups was analyzed by ANOVA, using Tukey’s multiple comparison test. Note: The asterisk (*) represents significant difference (*p* < 0.05) when compared with mock-infected and mock + FCPLJ 1000 mice. The hash (#) represents significant difference (*p* < 0.05) when compared with the infected mice. Each experimental group consists of five mice (*n* = 5).

**Figure 4 pathogens-10-00501-f004:**
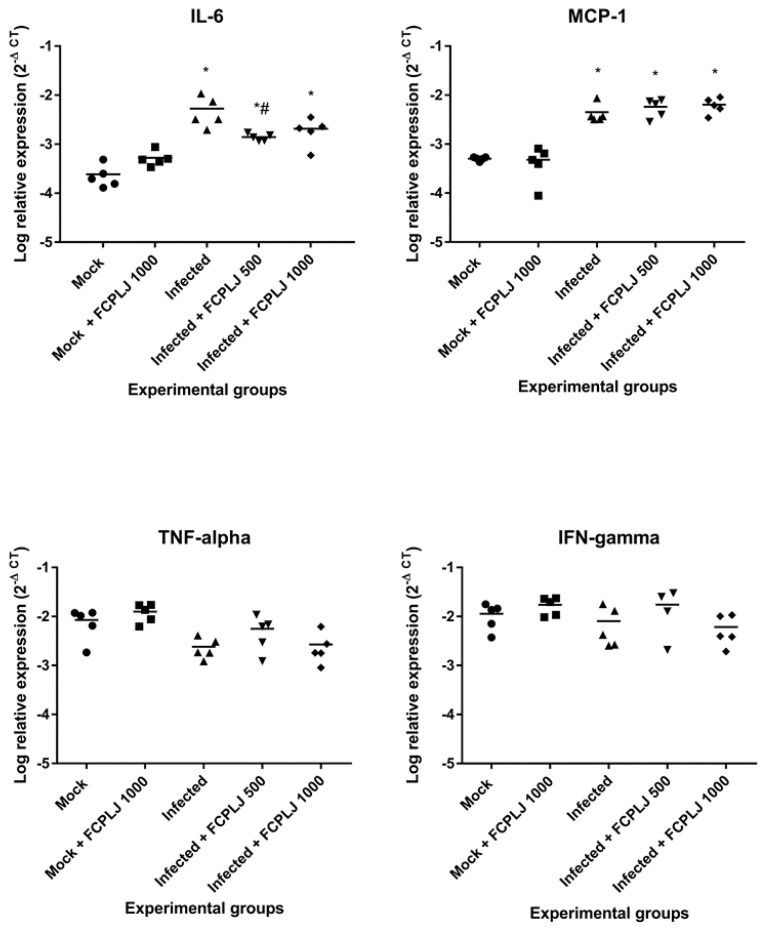
The level of intracellular cytokine expression in the liver. The expression level of four cytokines (MCP-1, IL6, IFN and TNF) in the liver tissues harvested on day four post-infection were determined by quantitative RT-PCR. The liver IL-6 expression level in the infected AG129 mice was higher than the mock infected AG129 mice. The FCPLJ treatments decreased the level of IL-6 expression in the liver tissue of infected mice. The comparisons between groups were analyzed by ANOVA, using Tukey’s multiple comparison test. Note: The asterisk (*) represents significant difference (*p* < 0.05) when compared with mock-infected and mock + FCPLJ 1000 mice. The hash (#) represents a significant difference (*p* < 0.05) when compared with the infected mice. Each experimental group consists of five mice (*n* = 5).

**Figure 5 pathogens-10-00501-f005:**
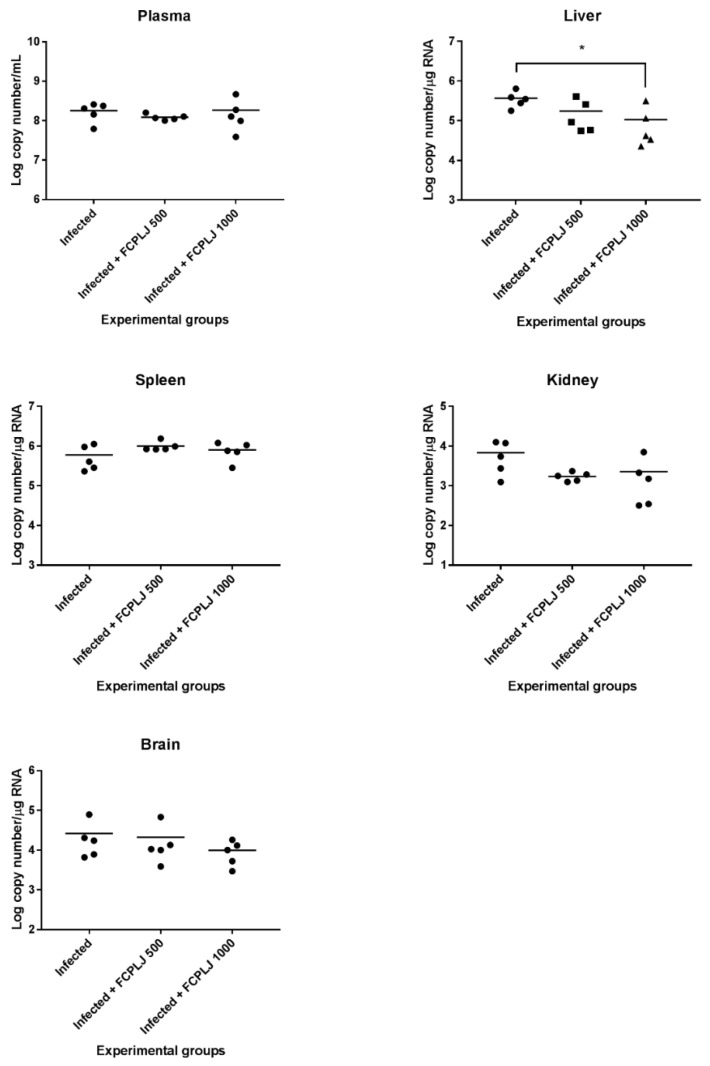
Dengue viral RNA level in the plasma and organs of experimental AG129 mice groups. The viral RNA copy numbers in plasma, liver, spleen, kidney and brain tissues harvested on day four post-infection were determined by quantitative RT- PCR technique. Except for the liver tissue, viral RNA level in the plasma and tissues of other organs was not affected by FCPLJ treatment. The comparison between groups was analyzed by ANOVA, using Tukey’s multiple comparison test. Note: The asterisk (*) represents a significant difference (*p* < 0.05). Each experimental group consists of five mice (*n* = 5).

**Table 1 pathogens-10-00501-t001:** Quantity of the freeze-dried *C. papaya* leaf juice (FCPLJ) main compounds.

Sample	Regression	Amount in FCPLJ (mg/g)				
		1	2	3	Average ^1^	%RSD ^1^
Manghaslin	R^2^ = 0.9927	5.95	5.10	6.10	5.71 ± 0.54	9.45
Clitorin	R^2^ = 0.9997	7.06	5.46	7.88	6.80 ± 1.23	18.11
Rutin	R^2^ = 0.9933	1.37	1.49	1.53	1.46 ± 0.08	5.56
Nicotiflorin	R^2^ = 0.9966	1.17	1.57	1.59	1.44 ± 0.24	16.61
Carpaine	R^2^ = 0.9989	4.06	3.43	3.97	3.82 ± 0.34	8.97

^1^ The average amount of compound was presented as mean ± standard deviation. RSD, Relative standard deviation.

## Data Availability

The raw data presented in this study are available on request from the corresponding author.

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
