# Peer review of "Immunomodulatory Activities of Carica papaya L. Leaf Juice in a Non-Lethal, Symptomatic Dengue Mouse Model"

_pathogens, 2021, doi:10.3390/pathogens10050501_

Round 1

Reviewer 1 Report

This manuscript written by Mohd Abd Razak et al. describes the immunomodulatory effect of Carica papaya L. leaf juice in AG129 mice infected with a clinical dengue virus isolate serotype 2. The authors show that the cell numbers of white blood cells and neutrophils significantly increase in C. papaya L. leaf juice-administered groups compared with in the control group and that oral administration of C. papaya L. leaf juice significantly decreases the plasma levels of GM-CSF, GRO-alpha, MIP-1 beta, IL-6, and MCP-1. In addition, they show that oral administration of C. papaya L. leaf juice significantly decreases viral RNA copy number in liver. The manuscript is well written and straightforward; however, there are several concerns to be addressed.

The authors need to discuss the reason why the oral administration of C. papaya L. leaf juice significantly decreases the plasma levels of GM-CSF, GRO-alpha, MIP-1 beta, IL-6, and MCP-1. In addition, why does C. papaya L. leaf juice significantly decrease viral RNA copy number in liver, but not in plasma, spleen, kidney, or brain? What is the potential action mechanism?

There are a number of papers reporting the effect of C. papaya L. leaf on dengue fever. This reviewer does not understand the significance of this study as compared with those papers. The authors need to clarify the significance of this study and to discuss the results obtained in this study by comparing with the data reported in other papers.

What can be the ingredient with the immunomodulatory effect in C. papaya L. leaf juice? As the authors mentioned in the text, the ingredients in C. papaya L. leaf are variable depending on where and when leaves are sampled. The variation in the content of ingredients often makes it difficult to reproduce the results shown in the papers reporting the effect of C. papaya L. leaf on dengue fever. The authors should discuss the bioactive ingredient in C. papaya L. leaf, which modulated the immune system of the dengue virus-infected AG129 mice.

Reviewer 2 Report

Interesting topic, data not as strong as I would like to be "sold" on the use of C. papaya juice for treating symptoms of DEN but it is a starting place.

Minor edits:

Throughout - consider renaming your DEN isolate to something such as - DENV-2 (DMOF15) or something else you feel is appropriate. 

Abstract infected with a clinical dengue (add "a")

Change to RT-qPCR

Introduction

Reword line 36 

37 tested, to date, were clinically

43 modulate immune

58 needs 

61 lacks interferon

63 remove The

65 leakage which leads

70 with a laboratory strain of dengue

Discussion

Remove The line 221

300 used in other studies 

Reviewer 3 Report

In this paper, Razak et. al, investigate the immunomodulatory activities of Carica papaya L. leaf juice using a clinical isolate of dengue virus 2 in a non-lethal, symptomatic dengue mouse model. The paper is very well written, thorough.

The manuscript in question is a well-written, straight forward descriptive study evaluating the ability of freeze-dried Carica papaya L. leaf juice (FCPLJ) to modulate the immune response in a symptomatic dengue disease mouse model. This is a follow-on study using a clinical dengue isolate rather than a lab adapted strain to establish the dengue mouse model. Using this model, FCPLJ treatment significantly decreased plasma levels of several proinflammatory cytokines and increased total white blood cell and neutrophil counts in the infected mice. These results support a potential role of FCPLJ as an immunomodulator in a symptomatic dengue mouse model. The authors satisfactorily acknowledge and address study limitations. They briefly touch on potential mechanisms/active ingredients that may explain the anti-inflammatory properties of C. papaya seen in this study, but are careful not to over speculate. Future studies should be conducted to determine the active ingredient(s) in FCPLJ and to elucidate the mechanism of action, but such investigations are beyond the scope of this manuscript as written. Other than a few minor typos, which can easily be corrected during proofing, I do not have any specific comments. 

Author Response

Point 1. Other than a few minor typos, which can easily be corrected during proofing, I do not have any specific comments. 

Response 1: Minor typo errors have been identified and corrected. 

1. 'Alergy' changed to 'Allergy'.

Page 1, Line 8.

2. 'Till' changed to 'Until'.

Page 13, Line 424.

Round 2

Reviewer 1 Report

The authors have addressed this reviewer's concerns well.